# Guaianolide Derivatives from the Invasive *Xanthium spinosum* L.: Evaluation of Their Allelopathic Potential

**DOI:** 10.3390/molecules27217297

**Published:** 2022-10-27

**Authors:** Sylvain Baldi, Pascale Bradesi, Alain Muselli

**Affiliations:** Université de Corse, UMR CNRS 6134 Sciences Pour l’Environnement, Laboratoire Chimie des Produits Naturels, BP 52, 20250 Corte, France

**Keywords:** *Xanthium spinosum*, essential oil, volatile metabolites, guaianolide, allelopathy

## Abstract

Ziniolide, xantholide B (11α-dihydroziniolide), and 11β-dihydroziniolide, three sesquiterpene lactones with 12,8-guaianolide skeletons, were identified as volatile metabolites from the roots of *Xanthium spinosum* L., an invasive plant harvested in Corsica. Essential oil, as well as hydrosol and hexane extracts, showed the presence of guaianolide analogues. The study highlights an analytical strategy involving column chromatography, GC-FID, GC-MS, NMR (1D and 2D), and the hemi-synthesis approach, to identify compounds with incomplete or even missing spectral data from the literature. Among them, we reported the ^1^H- and ^13^C-NMR data of 11β-dihydroziniolide, which was observed as a natural product for the first time. As secondary metabolites were frequently involved in the dynamic of the dispersion of weed species, the allelopathic effects of *X. spinosum* root’s volatile metabolites were assessed on seed germination and seedling growth (leek and radish). Essential oil, as well as hydrosol- and microwave-assisted extracts inhibited germination and seedling growth; root metabolite phytotoxicity was demonstrated. Nevertheless, the phytotoxicity of root metabolites was demonstrated with a more marked selectivity to the benefit of the monocotyledonous species compared to the dicotyledonous species. Ziniolide derivatives seem to be strongly involved in allelopathic interactions and could be the key to understanding the invasive mechanisms of weed.

## 1. Introduction

Invasive alien species represent one of the current main environmental issues; they are recognized by the Convention on Biological Diversity as the fourth cause of global biodiversity loss, after the disappearance of natural habitats, overexploitation of resources, and pollution [1]. Invasive species generally have rapid growth, high fertility, high dispersal power, and resistance to pathogens. Their numerous nuisance mechanisms (hybridization, modification of natural habitats, pathogenic organisms, etc.) allow them to take advantage of native species with the consequence of their disappearance and radical landscape modifications [2]. However, the chemistry of invasive species must vary, and with enormous biological activity potential that remains to be explored [3]. Invasive plants may serve as inexpensive and renewable sources of bioactive compounds.

The European Plant Protection Organization (E.P.P.O.) lists 6658 exotic species, of which 168 are considered invasive (78 in France) [4]. In Corsica, among the 2978 plant species listed, 454 introduced species have been recorded [5], of which, 52 are considered as worrying and 17 as invasive [6]. Among them, *Xanthium spinosum* (Spiny cocklebur) is a highly invasive plant originating from South America and is now widespread throughout the world [6,7,8,9,10,11,12]. *X. spinosum* is one of the worrying species because of its good adaptation to the Mediterranean climate as well as its affinity for nitrogenous soils. The plant is frequently found in farmlands where it causes a sanitary risk for cattle [8]. *X. spinosum* is provided with hooked spines, which can attach to animal coats and clothing, contributing to the dispersal over large areas.

*X. spinosum* is known by traditional medicine in many countries, such as Romania, Serbia, Egypt, South Africa, and Argentina [9]. The plant is used in the treatment of rabies, chronic fevers, and diabetes, and also to stimulate saliva production and for diuretic effects [10]. In Romania, plants are used to treat urinary problems and various prostate pathologies [11]. In Oltenia (southern Romania), *X. spinosum* and *X. italicum* seeds are used in infusion to help cardiac disorders [12]. In Bolivia, root decoctions are used to treat arteriosclerosis and hypertension, leaving decoction to inflammatory pathologies, such as oophoritis, hepatitis, toothache, cystitis, nephritis, and gastritis [13]. In North America, the Cherokee uses the plant to treat lung problems and snakebites [14]. Nowadays, the plant is used around the world in a wide range of medical applications. In Spain, the leaves are used as contraceptive drugs [15], and fruits are used to treat kidney malfunctions and hyperglycemia [16]. In Italy, a seed decoction is used to treat diarrhea [17].

Many studies have focused on the composition of polar solvent extracts and the chemical compounds identified are sorted into three main groups: phenolics, sesquiterpenes, and diterpenes [11,18,19,20,21,22,23,24,25,26,27]. Furthermore, *X. spinosum* is recognized for its biological and pharmaceutical activities, imputed to the presence of sesquiterpene lactones called xanthanolides [20,21,22,23,24,25,26,27,28,29,30,31,32,33,34,35,36,37,38,39,40,41,42,43,44]. These lactones, provided with a non-cyclic carbon chain and a seven-membered ring, appear to be responsible for cytotoxic [29,30,31], antitumor [20,31,32,33], antibacterial [34,35,36,37], anti-fungal [38], anti-leishmaniosis [38], anti-malarial [39], anti-inflammatory [30], and anti-ulcer activities [40,41]. In particular, xanthatin, isolated from the aerial parts of *X. spinosum*, is recognized for antibacterial and antifungal properties [37,42], as well as anti-angiogenesis [43] and phytotoxic properties [21,44].

Few studies relate to the chemical characterization of essential oils of the genus *Xanthium*, eleven describe the essential oil obtained from aerial parts of eight different species with various geographical origins: *X. cavanelesii* from Argentina [45], *X. canadense* from Japan [46], *X. sibiricum* from China [47], *X. brasilicum* from Iran [48], *X. pennsylvanicum* from Russia [49], *X. strumarium* from Brazil [50], Iran [51], India [52], and Egypt [53], and *X. italicum* [54] and *X. spinosum* [8] from Corsica.

To our knowledge, *X. spinosum* volatile root metabolites have never been investigated, and it might be interesting to study their involvement in the plant invasion mechanism. Indeed, through the exudation of a wide variety of compounds, roots have a critical ecological impact on soil and can inhibit the growth of competition plant species [55,56].

In this context, our project was interested in studying the ecological role of a Corsican invasive *Xanthium spinosum*, through the analysis of its volatile root metabolites and the evaluation of their allelopathic potential with the ambition to preserve biodiversity and provide a lasting response to the economic and ecological problems raised by invasive plant species.

## 2. Results and Discussion

### 2.1. Chemical Composition of X. spinosum Root Extracts

#### 2.1.1. Essential Oil and Hydrosol

Essential oil and hydrosol were prepared by hydrodistillation from *X. spinosum* roots. The integrated analysis of essential oil (EO) identified 49 components, which accounted for 92.8% of the total amount (Table 1). Chromatogram of the EO is available in Appendix A. Essential oil showed the same proportion of oxygenated and hydrocarbon compounds (46.4%). Hydrocarbons were mainly represented by sesquiterpenes (36.1%) while monoterpenes were weak (10.3%). Oxygenated sesquiterpenes amounted to 39.1%, and among them, sesquiterpene lactones were higher (22.3%). The main components were *α*-isocomene **32** (6.1%), β-elemene **33** (8.5%), neryl 2-methylbutyrate **53** (6.2%), carotol **55** (9.4%), and ziniolide **66** (19.3%). The structures of the main EO components were reported in Figure 1.

Hydrosol was treated by liquid–liquid extraction and the integrated analysis of hydrosol extract (HYD) identified 40 components that accounted for 83.5% of the total amount (Table 1). Chromatogram of the HYD is available in Appendix A. Relative to EO, hydrosol extract was exclusively composed of oxygenated compounds; monoterpenoids amounted to 13.3% and the number of sesquiterpene lactones reached 3 times higher than EO (61.6% vs. 22.3%, respectively). The main components of HYD were terpinen-4-ol **21** (4.9%), xantholide B **65** (15.0%), and ziniolide **66** (42.6%).

#### 2.1.2. Hexane Extracts

The integrated analysis of the cold maceration extract (MAC) and the assisted microwave extract (MAE) identified, respectively, 30 and 32 components, which accounted for 74.1 and 84.0% of the total amount (Table 1). Chromatograms of MAC and MAE extracts are available in Appendix A. Both hexane extracts showed relatively similar compositions, which do not greatly differ from EO concerning hydrocarbon sesquiterpenes. However, we should note that hydrocarbon monoterpenes and most polar compounds were missing, to the benefit of sesquiterpene lactones. The main components were germacrene D **41** (9.2 and 6.9%), xantholide B **65** (11.7 and 15.0%), and ziniolide **66** (25.2 and 30.4%) for MAC and MAE extracts, respectively.

### 2.2. Analytical Strategy Applied to Identify X. spinosum Sesquiterpenic Lactones

We should note that 76 components were identified by comparing their EI–MS and RI with those compiled in the laboratory–MS library and 13 components were identified by the perfect match against RIn from the literature and commercial MS libraries [57,58]. However, compounds **65**, **66**, and **67** were not indexed and their univocal identification required the development of an analytical strategy involving column chromatography (CC), NMR experiments, and a hemi-synthesis procedure. 

MAE extract was selected for its high proportion of compounds that remained unidentified after preliminary analysis, combined with a high extraction yield (0.23% against 0.04, 0.06, and 0.12% for EO, HYD, and MAC, respectively). Thus, two consecutive CC were carried out from the MAE extract: The first was to separate non-polar from polar components, and the second was performed on the polar fraction using a silica gel column impregnated with AgNO_3_. The 14 fractions obtained were analyzed by GC and GC-MS and GC chromatograms demonstrated that fractions 11 and 10 contained **65** (63.5%) and **66** (69.8%), respectively. Column chromatography resolution was not sufficient to obtain **67** with a convenient purity (7.9% in Fraction 12).

### 2.3. Contribution of MS and NMR to the Identification of Lactones from X. spinosum Roots

#### 2.3.1. Ziniolide **66**

Compound **66** gave a molecular peak at *m*/*z* 230, suggesting a C_15_H_18_O_2_ formula. EI-MS of **66** (Appendix A) comes with a base peak at *m*/*z* 91 and a peak at *m*/*z* 119, such as oxygenated sesquiterpenes. The MS spectrum of **66** showed a satisfactory concordance with sesquiterpene lactones recorded in our MS library, such as dehydrocostuslactone (score matching 50%).

The unequivocal identification of **66** as 3,10(14),11(13)-guaiatrien-12,8-olide, a guaiane-type sesquiterpene lactone commonly known as ziniolide was carried by alignment of ^13^C-NMR and ^1^H chemical shifts reported in the literature [28,59]. For completeness, the whole set of recorded NMR data (Appendix A) confirm the above structure. Experimental ^13^C-NMR chemical shifts were given in Table 2. Relative to the literature data [28], our ^13^C-NMR assignment differed for C-2, C-4, C-5, C-10, and C-11 (Figure 2). Concerning the C-2 and C-5 chemical shifts, the values given in the literature appear to be inverted. Correct assignments were carried out using APT experiments and the analysis of the long-rang correlations in the HMBC spectrum. The multiplicity of C-2 (δ_C_ 35.96) and C-5 (δ_C_ 51.83) as CH_2_ and CH, respectively, as well as the condensed five-membered ring system C1-C5, were clearly established. The chemical shift values of the three ethylenic quaternary carbons C-4, C-10, and C-11 were very close and a source of confusion. Their assignments were aided by HMBC: C-4 (δ_C_ 142.83) was correlated to H_3_-15, C-10 (δ_C_ 143.71) with H_2_-9 and H_2_-14, as well as C-11 (δ_C_ 141.40) with H_2_-13. In addition, the *cis*-stereochemistry of the bicyclo[5.3.0]decane junction and the γ-lactone arrangement of **66** were ensured by distinct differences with the ^13^C-NMR data of centaurolide-B [60], an analogous derivative of *trans*-*trans*-guaianolide.

Ziniolide was isolated for the first time from *Zinnia peruviana* (L.) L. (Asteraceae) (synonym *Zinnia multiflora*) [59], then also from *X. canadense* [46,61] and *X. catharticum* [28].

#### 2.3.2. Xantholide B (11α-dihydroziniolide) **65**

The EI-MS spectra of **65** and **66** were nearly the same except for heaviest fragments, such as molecular ions at *m*/*z* 230 and 232, respectively, which suggested that **65** was a dihydrogenated derivative of **66**. Moreover, **65** MS spectrum has a good concordance score with sesquiterpene lactones present in our MS libraries.

Despite the occurrence of ziniolide **66** (21.4%), ^13^C-NMR chemical shifts of **65** were easily extracted from the fraction 11 spectrum (Figure 3) according to their relative intensities (1:3, respectively) and fifteen resonances were isolated (Table 2). The ^13^C-NMR spectrum of **65** showed strong similarities with those of ziniolide **66** and indicated that this compound exhibited the same sesquiterpene skeleton with a vinyl methyl, only one exomethylene group, and a lactone functionality. The hypothesis of the occurrence of a dihydro lactone was supported by the comparison of both sets of chemical shifts between **65** and **66**, for which C-11, C-12, and C-13 showed the main differences. More precisely, the exomethylene group including C-11 (δ_C_ 141.40) and C-13 (δ_C_ 122.15) in **66** was replaced by a methine (C-11, δ_C_ 45.70) and a methyl (C-13, δ_C_ 15.74) groups, respectively, according to the hydrogenation of α-methylene-γ-lactone function. Relative to **66**, the carbonylic C-12 (δ_C_ 179.70) undergoes a strong shielding (9 ppm) due to the non-conjugated carbonyl system. This chemical shift value was in agreement with those of dihydro guaianolide [62]. In addition, chemical shifts of the methyl group (δ_C_ 15.74, δ_H_ 1.31) indicated an α orientation as reported in compounds with similar configurations [29]. Consequently, **65** was identified as the 11α-dihydroziniolide **66**. Finally, the agreement between ^1^H chemical shifts of **65** and those reported in the literature [46] supported the identification of 11α-dihydroziniolide, commonly known as xantholide B. It is to be noticed that xantholide B **65** has previously been isolated from *X. canadense* (L.) L. (Asteraceae) [46,61], and to our knowledge, the ^13^C-NMR data of **65** are described here for the first time. Recorded NMR data is given in Appendix A.

#### 2.3.3. 11β-dihydroziniolide **67**

We should note that component **67** has demonstrated peak overlapping with ziniolide **66** on our lab non-polar GC column (Rtx-1); the polar GC column (Rtx-wax) was required to obtain sufficient resolution (Appendix A). Compounds **67** and **65** exhibited identical EI-mass spectra, which suggest a diastereoisomeric relationship between both molecules (Figure 4). Therefore, **67** is also a dihydrogenated derivative of ziniolide **66**.

The reduction of a rich-ziniolide fraction (MAE-F10, **66**, 69.8%) using sodium borohydride (NaBH_4_) confirms our hypothesis. GC chromatogram and EI-MS of the NaBH_4_ reduction product have informed about the presence of a mixture of **65** (17.6%) and **67** (73.3%). The ultra-dominant abundance of **67** allowed for extracting the fifteen chemical shifts from the ^13^C-NMR spectra.

^13^C-NMR chemical shifts of **67** (Table 2) and more precisely the presence of a quaternary carbon C-12 (δ_C_ 179.03) confirmed the good efficiency of the reduction of α-methylene-γ-lactone function. In the same way, the substitution of the exocyclic methylene of **66** by a methyl (H_3_-13, δ_H_ 1.19, d 7.3 Hz) coupled to a methine (H-11, δ_H_ 2.88, q 7.3 Hz) confirmed the hydrogenation of the C_11_–C_13_ double bond.

The many similarities between the ^1^H and ^13^C-NMR data of **65** and **67** supported the hypothesis of a diastereoisomeric relationship emitted from the mass spectra. Most chemical shift variations between **65** and **67** were observed on C-6 (δ_C_ 29.74 and 22.67, respectively) and C-13 (δ_C_ 15,74 and 10.0, respectively), with reciprocal shielding γ effects resulting from the small dihedral angle between C-13 and C-6, which supposed β orientation of C-13 methyl group. Finally, the complete assignment was supported by HSQC and HMBC experiments (Table 2). Consequently, the structure of compound **67** was established as 11β-dihydroziniolide. The whole set of recorded NMR data is available in Appendix A.

The 11β-dihydroziniolide was previously reported as a reduction product of ziniolide and also as the reaction product of the treatment in basic conditions of xantholide B [46]. However, to our knowledge, the present work report for the first time the occurrence of 11β-dihydroziniolide as a natural product as well as the ^13^C-NMR data (Table 2) and ^1^H-NMR data (see experimental) was never published before. 

### 2.4. Allelopathic Effect of X. spinosum Root Extracts

Allelopathic activity of *X. spinosum* root extracts was assessed on the seed germination and seedling growth of two plants chosen for their respective botanical characteristics: (i) *Allium porrum* (*Alliaceae*) is an old, rustic, and perennial vegetable that we have selected as monocotyledon model and (ii) *Raphanus sativus* (*Brassicaceae*) is an annual or biennial plant, mainly cultivated for its fleshy and more popular hypocotyl, which was selected as dicotyledon model. The allelopathic potential was evaluated via the phytotoxicity of three root extracts: EO, HYD, and MAE-F10; a rich-lactone fraction obtained by CC from MAE. All extracts were assessed at concentration ranges from 0 to 1000 µg/mL. The phytotoxicity was appreciated through biological indices: root and shoot lengths, wet and dry weights, germination rate, germination percentage, vigor index, length and weight ratios, and allelopathic effect (see the experimental section for further details). The results were reported in Table 3 for *A. porrum* and *R. sativus*, respectively.

Our experiments highlighted the inhibitory effect of *X. spinosum* root extracts on the growth of both species. As seen in Figure 5a, the growth of the monocotyledon *A. porrum* seeds was strongly disturbed in contact with *X. spinosum* extracts. Relative to the control, the inhibition growth was clearly correlated to the concentration increase of the three natural extracts. Concerning *R. sativus* seeds (Figure 5b), the growth inhibition was disturbed to a lesser extent. The dicot seeds of *R. sativus* appear less sensitive than the monocot seeds of *A. porrum* to the toxic effects of *X. spinosum* extracts. Nevertheless, used at a high concentration (1000 μg/ml), the hydrosol extract was the most efficient to inhibit the seedling lengths of *R. sativus*.

Analysis of variance on the mean of seedling lengths (L) (Table 3) indicated that monocotyledon L was significantly different from control for all extracts (no treatment showed a common letter with control). While for the dicot plant, L was significantly different from the control only with hydrosol extract and MAE-F10 at high concentrations (from 250 to 1000 µg/mL).

The effect of *X. spinosum* natural mixtures on the seedling length could be analyzed using the length ratio (LR). While the root and shoot growths were little differentiated for the monocot seeds of *A. porrum* without treatment, a higher concentration of lactone-rich extract (MAE-F10) appeared to improve the specific growth of aerial shoots to the detriment of the roots. Concerning the seeds of dicot *R. sativum*, the root growth was twice higher than aerial shoot growth without treatment. Our experiment highlighted an increase of length ratio (LR) when the treatment concentration increased, whatever the *X. spinosum* extract used. This indicated that dicot radicles were more sensitive to extracts than the hypocotyl.

Relative to control, the treatment of the monocot seeds of *A. porrum* by an increasing concentration of *X. spinosum* extracts clearly affected the germination rate (GR). In addition, most seeds did not germinate. For completeness, a germination percentage (GP) followed the same tendency, the increase of extract concentrations caused a lack of germination success. The MAE-F10 extract at 1000 µg/mL was the more efficient, only a third of *A. porrum* seeds reached the germination state at the end of the experiment. Concerning dicot seeds of *R. sativus*, whatever the concentration, GR was less affected by *X. spinosum* extracts, and GP was not affected by treatments.

Comparison of vigor index (VI) and the allelopathic effect (AE) of both species confirmed that monocot seedling growth was more affected by the root extracts than the dicot seedlings; VI of the dicot plant was slightly affected whereas the VI of the monocot plant was from 8 to 45 times smaller at higher extract concentrations.

Treated monocot seeds that have successfully grown appeared to be significantly smaller than the control and a negative allelopathic effect (AE) was observed for all treatments. A greater effect was observed with MAE-F10 extract with an AE of −95.7% at 1000 µg/mL. Dry weight and wet weight were both uniformly affected, as shown by the constant weight ratio (WR), suggesting that the inhibition of seedling growth could be attributed to the inhibition of mitosis (increase in biomass). We should note that HYD extract treatment stimulated seedling growth at 100 µg/mL, as shown by the positive AE, a phenomenon known as “low dose simulation–high dose stimulation” or “hormesis” [63].

Sesquiterpene lactones are recognized for their allelopathic potential thanks to their α,β-unsaturated carbonyl group, which can undergo 1,4-conjugated additions with nucleophiles, such as the sulfhydryl group very abundant in proteins and nucleic acid [64]. Moreover, the biological effect of ziniolide **66** has already been the object of various studies and demonstrated larval growth inhibition of *Drosophila melanogaster* [46] as well as antibacterial activity [28], anti-inflammatory, cytotoxic, and antitumor potential [42].

Our biological assessments underlined the lactones responsibility in the phytotoxicity of *X. spinosum* extracts and the α-methylene-γ-lactone group present in ziniolide **66** could be the main actor. Thus, guaianolide in *X. spinosum* roots and their recent introduction into the island territory might be one of the factors encouraging the invasion of the species; as supposed by the “novel weapon hypothesis”, natives species may not yet have the time to develop metabolic pathways to eliminate and counteract these new phytotoxins [65].

However, the phytotoxic effect was not proportional to lactone content and lactones do not seem to be the only ones involved. Indeed, monoterpenes, such as pinene isomers [66], possess allelopathic properties, hydrocarbon monoterpenes contained in *X. spinosum* essential oil could explain the difference in the inhibition mechanism and the overall effect probably results from the positive or negative synergy between several metabolites.

## 3. Materials and Methods

### 3.1. Plant Material

*X. spinosum* roots were harvested in central Corsica (Corte, France, 42°17’56.5”N, 9°10’13.0”E) during a dormant state in January 2019. The botanical determination of the plants was performed according to the botanical keys summarized in Flora Corsica [5].

### 3.2. Isolation of Volatile Metabolites

Four sample preparation techniques including hydrodistillation, hydrosol extraction, cold maceration in hexane, and microwave-assisted extraction in hexane were used in order to produce exhaustive volatile extracts.

Air-dried roots (200 g) were subjected to hydrodistillation (5 h) using a Clevenger-type apparatus, according to the method recommended in the *European pharmacopoeia* [67]. Hydrodistillation produced a yellow essential oil (EO) with a yield of 0.04% (w/*dw*, based on the weight of the dried plant material) and aromatic water, called hydrosol.

Hydrosol was recovered by removing co-coating (the first 300 mL) of the Clevenger apparatus during the hydrodistillation. Then, hydrosol was submitted to liquid–liquid extraction (LLE). A total of 300 mL was extracted successively with 3 × 50 mL of diethyl ether; the organic phase was then washed with 50 mL of water saturated with NaCl, dried over Na_2_SO_4_, and filtered before being concentrated to produce hydrosol extract (HYD). Hydrosol extraction produced a colorless extract with a yield of 0.06% (w/*dw*).

Ground air-dried roots (20 g) were subjected to maceration in hexane (200 mL) at room temperature (48 h). The solvent was then filtered and concentrated. The resulting extract was next taken up in absolute ethanol and centrifuged (20 min at 6000 rpm), and the supernatant was collected and concentrated to finally obtain the macerate extract (MAC). Maceration in hexane produced an orange–yellow extract with yields of 0.12% (w/*dw*).

Air-dried roots were extracted using Multiwave 3000 (Anton Paar, Gratz, Austria) apparatus provided with 16 ceramic vessels. For each vessel, ground roots (5 g) were introduced with hexane (40 mL) and extraction was realized at 180 °C (150 W per vessel) for 20 min followed by 40 min of cooling. The solvent was then filtered and concentrated. The resulting extract was next taken up in absolute ethanol and centrifuged (20 min at 6000 rpm), and the supernatant was collected and concentrated to finally obtain the microwave extract (MAE). The microwave-assisted extraction produced an orange–yellow extract with a yield of 0.23% (w/*dw*).

### 3.3. Fractions

The MAE extract (1.44 g) was first submitted to column chromatography on a silica gel column (40 × 2 cm, 63–200 µm, 50 g) with two elution gradients (100/0 and 0/100 of Hex/DIPE) giving two fractions. The polar fraction (1.15 g) was next submitted to column chromatography on a silica gel column impregnated with 10% AgNO_3_ (40 × 2 cm, 40–63 µm, 50 g) using gradients of Hex/DIPE. TLC fingerprint grouping gave 14 fractions, among them, F10 (200 mg, **65**: 10.1%, **66**: 69.8%, **67**: 2.8%), F11 (75 mg, **65**: 63.5%, **66**: 21.4%, **67**: 4.3%) and F12 (9 mg, **65**: 16.4%, **66**: 6.8%, **67**: 7.9%).

### 3.4. NaBH_4_ Reduction

β-dihydroziniolide **67** was obtained by treating the MAE fraction 10 (20 mg, 69.8% of ziniolide **66**) with the NaBH_4_ solution (1.23.10^−4^ mol) in ethanol (10 mL). The resulting mixture was stirred at room temperature and then refluxed for 60 min. After treatment, 10 mL of water saturated with NaCl and 2 drops of glacial acetic acid were added. The mixture was then extracted by 3 × 15 mL of hexane and dried over sodium sulfate before being concentrated under a vacuum. The resulting mixture (12 mg) contained xantholide B **65** (17.6%) and β-dihydroziniolide **67** (73.3%).

### 3.5. GC-FID Analysis

Analyses were carried out using a Perkin–Elmer Clarus 600 gas chromatography (GC) apparatus (Waltham, MA, USA) equipped with a single injector and two flame ionization detectors (FIDs) for simultaneous sampling to two fused–silica capillary columns (60 m × 0,22 mm i.d., film thickness 0.25 μm; Restek, Bellefonte, PA, USA) with stationary phases of different polarity, i.e., a nonpolar Rtx–1 (polydimethylsiloxane) and a polar Rtx–Wax (polyethylene glycol). The oven temperature was programmed to increase from 60 to 230 °C at 2 °C min^−1^ and was held isothermal at 230 °C for 30 min. The injector temperature was maintained at 280 °C and the detector temperature at 280 °C, the carrier gas was H_2_ (0,7 mL.min^−1^) and the samples were injected (0.2 μL of pure oil) in the split mode (1:80). Retention indices (RIs) of the mixture components were determined relative to the retention times (t_R_) of a series of n-alkanes (C_5_–C_30_; commercial solution, obtained from Restek, Bellefonte, PA, USA) using the Van den Dool and Kratz equation [68].

### 3.6. GC-MS Analysis

The plant extracts and the fractions obtained by CC were investigated using a Perkin Elmer Turbo Mass quadrupole detector directly coupled to a Perkin Elmer SQ8 (Walton, MA, USA), equipped with the two same fused-silica capillary columns as described above. Both columns were used with the same quadrupole MS detector. The analyses were consecutively carried out on the nonpolar and the polar column. Hence, for each sample, two reconstructed ion chromatograms (RIC) were provided, which were investigated consecutively. The GC conditions were the same as described above and the MS parameters were as follows: ion–source temperature, 150 °C, ionization energy, 70 eV; electron ionization mass spectra acquired over a mass range of 35–350 amu during a scan time 1 s. The injection volumes were 0.1 μL.

### 3.7. NMR Analysis

Nuclear magnetic resonance (NMR) spectra were recorded on MAE fractions 10 and 11, and the NaBH_4_ reduction product of MAE fraction 10.

NMR experiments were acquired in CDCl_3_ (EuroIsotop, Saint Aubin, France), at 300 K using a Bruker Avance DRX 500 NMR spectrometer (Karlsruhe, Germany) operating at 500.13 MHz for ^1^H and 125.77 MHz for ^13^C Larmor frequency with a double resonance broadband fluorine observe (BBFO) 5 mm probe head. ^13^C-NMR experiments were recorded using a one–pulse excitation pulse sequence (90° excitation pulse) with ^1^H decoupling during signal acquisition (performed with WALTZ–16); the relaxation delay has been set at 2 s. For each analyzed sample, depending on the compound concentration, 3 up to 5 k free induction decay (FID) 64 k complex data points were collected using a spectral width of 30,000 Hz (240 ppm). Chemical shifts (δ in ppm) were reported relative to the residual signal of CDCl_3_ (δ_C_ 77.04 ppm). Complete ^1^H and ^13^C assignments of the new compound were obtained using 2D gradient–selected NMR experiments, ^1^H–^1^H COSY (correlation spectroscopy), ^1^H–^1^3C HSQC (heteronuclear single quantum correlation), ^1^H–^13^C HMBC (heteronuclear multiple bond coherence) and ^1^H–^1^H NOESY (nuclear Overhauser effect spectroscopy), for which conventional acquisition parameters were used, as described in the literature [69].

### 3.8. Compound Identification and Quantification

Identification of individual components in plant extracts or CC fractions was based on a methodology involving integrated techniques, such as GC retention indices, GC-MS (EI), and NMR. The identification of individual components was based (i) on the comparison of the retention indices (RIs) determined on the polar and nonpolar columns with those of authentic compounds or literature data [57,58]; (ii) on computer matching of the mass spectra with commercial MS libraries and the mass spectra with those listed in our homemade MS library of authentic compounds or literature data [57,58,70,71]; (iii) comparing the ^13^C-NMR chemical shifts of CC fraction components with those of reference spectra reported in the literature; (iv) NMR assignments using 1D and 2D experiments. The relative quantification percentage was obtained by internal normalization of the GC-FID peak area without response factors.

#### 11β-dihydroziniolide (**67**)

Colorless oil; *Rf* 0.14 (Hex/DIPE 6:4), 0.67 (Hex/Ethyl acetate 6:4); *MS m/z* 232 see Figure 4a; ^1^H NMR data (CDCl_3_, 600 MHz): δ 5.37 (1H, br s, H-3), δ 5.01 (1H, s, H-14a), δ 4.97 (1H, s, H-14b), δ 4.56 (1H, dt, *J* = 4.6, 11.2 Hz, H-8), δ 3.19 (1H, m, H-1), δ 2.88 (1H, q, *J* = 7.3 Hz, H-11), δ 2.82 (1H, dd, *J* = 6.8, 13.7 Hz, H-9b), δ 2.58 (1H, dd, *J* = 4.2, 13.7 Hz, H-9a), δ 2.41 (1H, br m, H-7), δ 2.40 (1H, br m, H-2a), δ 2.38 (1H, br m, H-5), δ 2.31 (1H, br m, H-2b), δ 1.74 (1H, s, H-15), δ 1.42 (1H, d, *J* = 13.2 Hz, H-6b), δ 1.19 (3H, d, *J* = 7.3 Hz, H-13), δ 1.17 (1H, s, H-6a); ^13^C (CDCl_3_, 125 MHz) see Table 2.

### 3.9. Allelopathic Effect Evaluation

The allelopathic activity was assessed on three *X. spinosum* root extracts (EO, HYD, and the MAE fraction 10) selected according to their chemical compositions. Allelopathic tests were performed using the methodology reported in the literature [72] and implemented in our laboratory.

Commercial seeds of *Allium porrum* (monocotyledon) and *Raphanus sativus* (dicotyledon) were used to assess the phytotoxicity of *X. spinosum* extracts. Stock solutions of essential oil were prepared in dimethylsulfoxide (DMSO) as the initial solvent followed by dilution with distilled water to a final concentration of 1000 μg·mL^−1^. The concentration of DMSO in the stock solution was 1% *v*/*v*. Other test solutions (100, 250, and 500 μg·mL^−1^) were prepared by dilution of the stock solution with distilled water. Control treatment (0) was an aqueous solution of DMSO (1% *v*/*v*). Three replicates, each of 10 seeds, were prepared for each treatment using glass Petri dishes (9 cm) lined with Whatman no. 4 filter paper. A total of 3 mL of test solution was added to each Petri dish. The Petri dishes were hermetically closed with stretch film and placed in an incubator at 20 °C in the dark. Germinated seeds were counted each day over a period of 7 days and root length, shoot length and wet seedling weight were determined after 7 days. Finally, germinated seeds were kept in a laboratory oven at 60 °C for one week to determine dry seedling weight.

At the end of the experiment germination rate (GR) [66,73], germination percentage (GP) [74], vigor index (VI) [75], lengths ratio (LR), weights ratio (WR) [66], and allelopathic effect (AE) [76] were determined from the following equations:(1)$$\mathrm{GR}=\sum_{i=1}^{n}\frac{ni}{di}\;$$
(2)$$\mathrm{GP}\;=\frac{nf}{N}\times 100$$
(3)$$\mathrm{VI}\;=\frac{\mathrm{GP}\;\times \;L}{100}$$
(4)$$\mathrm{LR}\;=\frac{S}{R}$$
(5)$$\mathrm{WR}\;=\frac{dw}{\mathrm{ww}}$$
(6)$$AE=\left(\frac{L}{C}-1\right)\times 100$$
n*i*: number of germinated seeds at each counting; d*i*: number of days until x counting; x: counting number; n*f*: number of germinated seeds at the end of the experiment; N: total number of seeds; L: mean of seedling lengths (mm); *dw*: mean of seedling dry weights (mg); ww: mean of seedling wet weights (mg); R: mean of root lengths (mm); S: mean of shoot lengths (mm); *C*: mean of control seedling lengths (mm).

### 3.10. Statistical Analysis

Seedling length data from allelopathic bioassays were subjected to one-way analysis of variance (ANOVA) using Minitab statistical software. Means of multiple treatments were compared using Tukey HSD (honestly significant difference) test at a 5% level of significance. All results are expressed as mean ± SD. Means not sharing a common letter are significantly different.

## 4. Conclusions

The volatiles of *X. spinosum* roots were characterized by an analytical strategy involving column chromatography, GC-FID, GC/MS, NMR, as well as hemi-synthesis to identify guaianolide sesquiterpenes with incomplete or even missing spectral data from the literature. Instead of the xanthanolides usually found in the species, ziniolide, xantholide B (11α-dihydroziniolide), and 11β-dihydroziniolide, three sesquiterpene lactones with 12,8-guaianolide skeleton were identified from the essential oil, hydrosol extract, and hexane extracts from *Xanthium spinosum* L. Among them, ^1^H- and ^13^C-NMR data of 11β-dihydroziniolide, as well as its occurrence as a natural product were described for the first time. Our study aimed to highlight the involvement of volatile compounds of *X. spinosum* roots in the allelopathic interactions between these invasive weeds and plants. Treatments inflicted on leek and radish seeds involved essential oil, as well as hydrosol and lactone-rich extracts of *X. spinosum* roots, which have shown phytotoxicity. *Allium porrum*, chosen as a monocot model, appears more sensitive than the dicot *Raphanus sativus* concerning seed germination. Nevertheless, used at a high concentration (1000 μg/ml), the hydrosol extract was the most efficient to inhibit the seedling length of *R. sativus* and selective inhibition of radicle seedlings was observed according to the concentration treatment. The present study demonstrated great seedling growth inhibition and the anti-germination potential of *X. spinosum*. The involvement of the ziniolide analogs appears to be effective; nevertheless, the aid of other natural products in the extracts can contribute to synergetic effects. In order to develop valorization opportunities for *X. spinosum*, further investigations were required to determine its possible use as a natural herbicide.

## Figures and Tables

**Figure 1 molecules-27-07297-f001:**
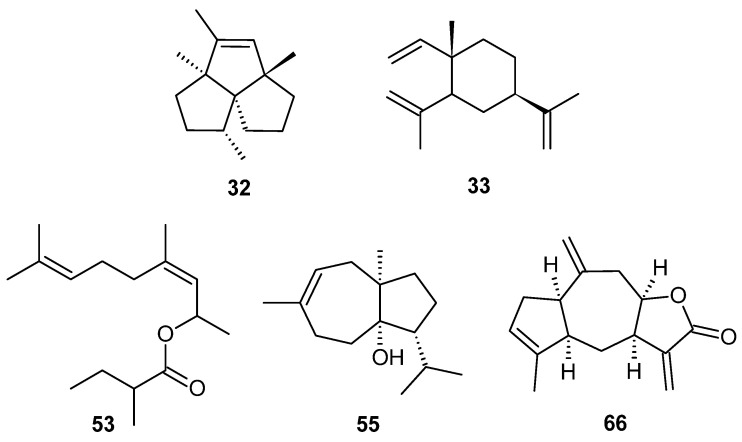
Structures of main components of *X. spinosum* root essential oil from Corsica; α-isocomene **32**, β-elemene **33**, neryl 2-methylbutyrate **53**, carotol **55**, ziniolide **66**.

**Figure 2 molecules-27-07297-f002:**
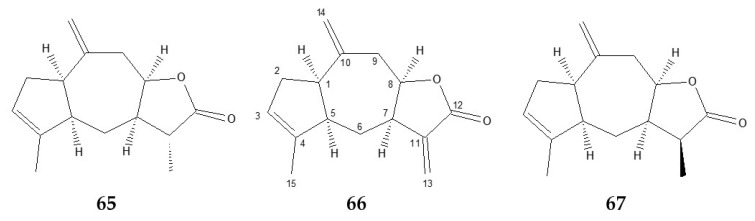
Structures of xantholide B **65**, ziniolide **66**, and 11β-dihydroziniolide **67**.

**Figure 3 molecules-27-07297-f003:**
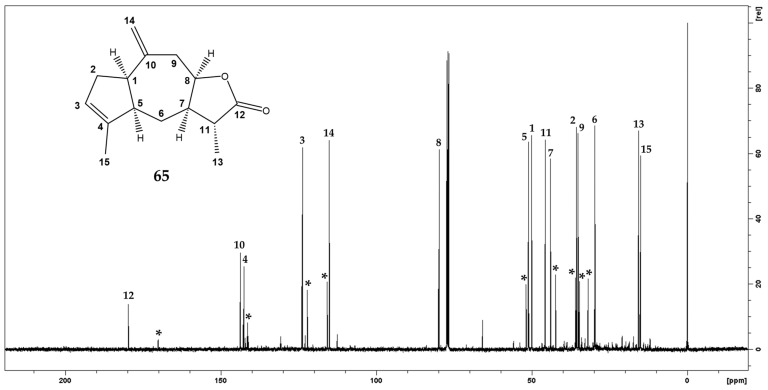
NMR ^13^C spectrum of the MAE fraction 11 in CDCl_3_ (125.77 MHz, at 300 K) in which two compounds are evidenced: xantholide B **65** (63.5%)—signal assignments refer to numbering in the inserted structure; and ziniolide **66** (21.4%)—signals marked with (*****).

**Figure 4 molecules-27-07297-f004:**
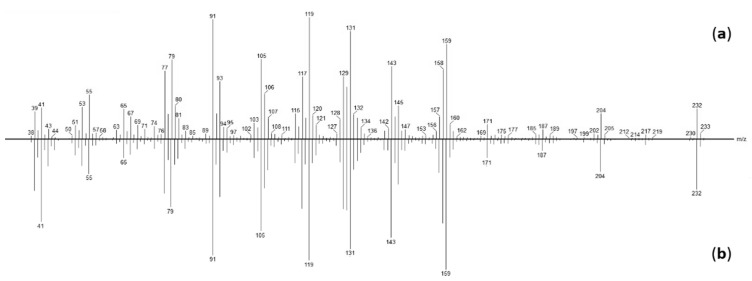
EI-MS spectrum of (**a)** compound **67**; (**b**) xantholide B **65**.

**Figure 5 molecules-27-07297-f005:**
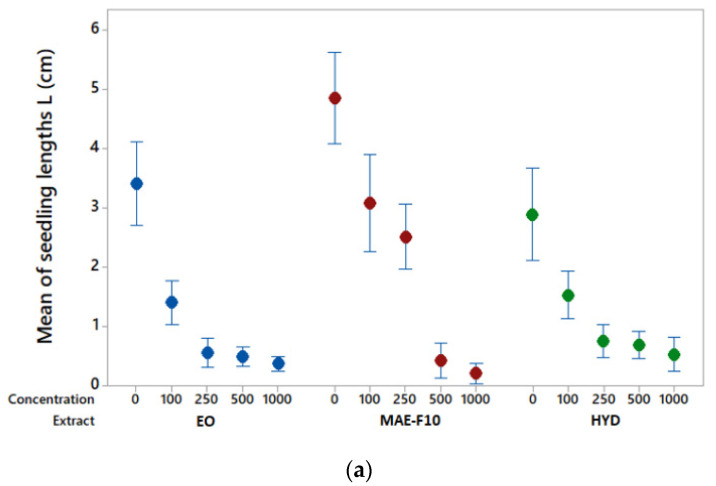
Effect of *X. spinosum* extracts on seed growth. Data displayed is the mean of three replicates of 10 seeds each, individual standard deviations were used to calculate the intervals. Mean of seedling lengths (L) were measured after an incubation time of 7 days at 20 °C in the dark using seeds of (**a**) *Allium porrum* (monocot plant) and (**b**) *Raphanus sativus* (dicot plant). Seeds were treated using five extract concentrations from 0 (control) to 1000 μg.mL^−1^. EO: essential oil, HYD: hydrosol extract, MAE-F10: rich-lactone fraction obtained by CC from MAE.

**Table 1 molecules-27-07297-t001:** Chemical compositions of *X. spinosum* roots.

N° ^a^	Constituents	RIn_lit_ ^b^	RIn_exp_ ^c^	RIp_exp_ ^d^		Content (%) ^e^					Identification ^f^
						EO	HYD	MAC	MAE		
1	Hexanal	801	770	1055		-	0.1	-	-		RI, MS
2	Benzaldehyde	941	929	1525		-	0.2	-	-		RI, MS
3	Tricyclene	927	920	1020		0.1	-	-	-		RI, MS
4	α-Thujene	932	928	1023		0.2	-	-	-		RI, MS
5	α-Pinene	936	931	1022		1.5	-	-	-		RI, MS
6	Camphene	950	943	1066		1.9	-	-	-		RI, MS
7	β-Pinene	978	970	1110		1.9	-	-	-		RI, MS
8	α-Terpinene	1013	1008	1178		2.0	-	-	-		RI, MS
9	p-Cymene	1015	1011	1268		1.1	-	-	-		RI, MS
10	Limonene	1025	1020	1199		0.8	-	-	-		RI, MS
11	γ-Terpinene	1051	1047	1243		0.6	-	-	-		RI, MS
12	Terpinolene	1082	1078	1280		0.2	-	-	-		RI, MS
13	*cis*-Sabinene hydrate	1048	1051	1605		-	0.5	-	-		RI, MS
14	*trans*-Sabinene hydrate	1082	1083	1541		-	0.6	-	-		RI, MS
15	*cis*-p-Menth-2-ene-1-ol	1108	1106	1621		-	0.3	-	-		RI, MS
16	Camphor	1123	1121	1522		-	0.1	-	-		RI, MS
17	*trans*-p-Menth-2-ene-1-ol	1123	1122	1606		-	0.2	-	-		RI, MS
18	*trans*-Verbenol	1140	1129	1676		-	0.1	-	-		RI, MS
19	Mentha-1,5-dien-8-ol	1166	1145	1698		-	0.1	-	-		RI, MS
20	Borneol	1150	1148	1698		-	1.6	-	-		RI, MS
21	Terpinen-4-ol	1164	1161	1600		1,0	4.9	-	-		RI, MS
22	α-Terpineol	1176	1179	1700		tr	0.6	-	-		RI, MS
23	Cosmen-2-ol	1187	1198	1824		-	3.9	-	-		RI, MS
24	Nerol	1210	1211	1799		-	0.3	-	-		RI, MS
25	Geraniol	1235	1244	1731		tr	0.1	-	-		RI, MS
26	7α-Silphiperfol-5-ene	1329	1328	1429		-	-	0.1	-		RI, MS
27	Silphin-1-ene	1350	1348	1474		1.2	-	0.5	0.4		RI, MS
28	Cyclosativene	1378	1376	1483		0.2	-	-	0.1		RI, MS
29	Daucene	1380	1382	1502		tr	-	0.1	0.6		RI, MS
30	α-Copaene	1379	1379	1488		1.3	-	0.3	0.5		RI, MS
31	Modhephene	1383	1382	1522		1.4	-	0.8	0.6		RI, MS
32	α-Isocomene	1389	1388	1533		6.1	-	4.2	3.7		RI, MS
33	β-Elemene	1389	1388	1589		8.5	tr	2.4	4.2		RI, MS
34	β-Isocomene	1411	1406	1571		2.0	tr	1.1	0.9		RI, MS
35	(E)-Caryophyllene	1421	1424	1591		0.5	-	0.5	0.5		RI, MS
36	γ-Elemene	1429	1429	1638		0.1	-	0.2	-		RI, MS
37	*trans*-α-Bergamotene	1434	1432	1580		0.5	-	0.4	0.4		RI, MS
38	α-Humulene	1455	1456	1665		3.6	tr	1.6	1.2		RI, MS
39	4,5-di-epi-Aristocholene	1471	1467	1665		0.1	-	-	-		RI, MS
40	γ-Muurolene	1474	1471	1681		-	-	0.1	-		RI, MS
41	Germacrene-D	1479	1480	1704		1.1	tr	9.2	6.9		RI, MS
42	β-Selinene	1486	1483	1712		3.4	tr	0.4	0.4		RI, MS
43	α-Selinene	1494	1495	1720		3.1	-	0.4	0.5		RI, MS
44	α-Bulnesene	1503	1502	1711		-	-	-	0.3		RI, MS
45	β-Bisabolene	1503	1509	1744		0.4	-	0.4	0.3		RI, MS
46	γ-Cadinene	1507	1507	1752		0.3	-	-	0.6		RI, MS
47	δ-Cadinene	1520	1516	1752		1.6	-	0.3	tr		RI, MS
48	*trans*-Cadina-1,4-diene	1523	1523	1763		0.2	-	-	-		RI, MS
49	α-Calacorene	1527	1531	1895		0.3	-	-	-		RI, MS
50	β-Calacorene	1541	1548	1939		0.2	-	-	-		RI, MS
51	Spathulenol	1572	1568	2125		0.5	0.2	-	-		RI, MS
52	4-Formyl-5-nor-β-caryophyllene	1568	1564	1994		-	0.1	-	-		RI, MS
53	Neryl 2-methylbutyrate	1570	1565	1865		6.2	tr	2.3	1.4		RI, MS
54	Caryophyllene oxyde	1578	1576	1980		0.2	0.1	0.1	-		RI, MS
55	Carotol	1594	1594	2018		9.4	1.5	5.0	2.6		RI, MS
56	Humulene epoxyde II	1602	1601	2044		1,0	0.5	0.5	0.6		RI, MS
57	epi-Cubenol	1623	1640	2059		-	0.3	-	-		RI, MS
58	α-Cadinol	1643	1645	2231		1.3	-	0.2	-		RI, MS
59	*t*-Muurolol	1633	1634	2143		0.2	tr	0.8	-		RI, MS
60	Selin-11-en-4-α-ol	1658	1659	2231		3.1	2.3	0.7	3.7		RI, MS
61	Bulnesol	1665	1659	2204		tr	0.2	-	1.0		RI, MS
62	14-Hydroxy-9-epi-(E)-caryophyllene	1668	1657	2316		-	0.7	-	-		RI, MS
63	Eudesma-4(15),7-dien-1-β-ol	1671	1672	2347		-	1.2	-	-		RI, MS
64	14-hydroxy-α-Muurolene	1779	1758	2531		1.1	0.9	0.3	-		RI, MS
65	Xantholide B (11-α-dihydroziniolide)	-	1896	2785		3.0	15.0	11.7	15.0		RI, MS, NMR
66	Ziniolide (Xantholide A)	-	1921	2853		19.3	42.6	25.2	30.4		RI, MS, NMR
67	11-β-dihydroziniolide	-	1925	2838		tr	2.1	1.8	-		RI, MS, NMR
68	Hexadecenoic acid	1951	1951	2870		0.1	tr	2.4	1.2		RI, MS
69	Dihydrocollumellarin	1900	1956			-	0.9	-	-		RI, MS
70	Collumellarin	1952	1958	2891		-	1.0	-	6.0		RI, MS
	**Total identified**					**92.8**	**83.5**	**74.1**	**84.0**		
	Hydrocarbon compounds					46.4	tr	23.05	22.1		
	Oxygenated compounds					46.4	83.5	51.0	61.9		
	Hydrocarbon monoterpenes					10.3	-	-	-		
	Hydrocarbon sesquiterpenes					36.1	tr	23.05	22.1		
	Oxygenated monoterpenes					7.2	13.3	2.3	1.4		
	Oxygenated sesquiterpenes					39.1	69.9	46.3	59.3		
	Other oxygenated compounds				0.1	0.3	2.4	1.2			
	Sesquiterpenic lactones					22.3	61.6	38.7	51.4		

^a^ Compounds are listed in order of their elution from non-polar Rtx-1 column. ^b^ RIn_lit_: retention indices for a non-polar column taken from Konïg et al. [57], and from Adams et al. [58]. ^c^ RIn_exp_: Retention indices determined experimentally on the non-polar Rtx-1 column. ^d^ RIp_exp_: retention indices determined experimentally on the polar Rtx-wax column. ^e^ The contents (normalized abundances) were determined on the non-polar column Rtx-1 column; tr, trace (< 0.1%); EO: essential oil, HYD: hydrosol extract, MAC: cold-macerate in hexane, MAE: microwave-assisted extraction in hexane. ^f^ Identification methods: RI, comparison with retention indices; MS, comparison of mass spectra with those listed in mass-spectral libraries; NMR, structural elucidation via chemical shifts assignment. For details, see Experimental Section.

**Table 2 molecules-27-07297-t002:** ^13^C-NMR data ^a,b^ in CDCl_3_ for xantholide B **65**, ziniolide **66**, and 11β-dihydroziniolide **67**.

C *	65 δ_C_, type	66 δ_C_, type	HMBC	67 δ_C_, type	HMBC
1	50.10, CH	51.01, CH	3, 14	50.68, CH	2, 3, 14
2	35.67, CH_2_	35.96, CH_2_	1, 3	35.83, CH_2_	1, 3
3	123.75, CH	123.98, CH	1, 2, 15	123.84, CH	2, 15
4	142.55, C	142.83, C	2, 15	142.70, C	2, 15
5	51.13, CH	51.83, CH	1, 6, 15	50.69, CH	1, 6, 15
6	29.74, CH_2_	31.96, CH_2_	1, 7, 8	22.67, CH_2_	1, 11
7	44.00, CH	42.37, CH	8, 9, 13	43.75, CH	6, 9, 11
8	79.77, CH	80.02, CH	9	79.61, CH	6, 9
9	35.15, CH_2_	34.76, CH_2_	8, 14	34.96, CH_2_	14
10	143.68, C	143.71, C	1, 9, 14	142.56, C	1, 9, 14
11	45.70, CH	141.40, C	7, 13	40.57, CH	6
12	179.70, C	170.13, C	13	179.03, C	11, 13
13	15.74, CH_3_	122.15, CH_2_	7	10.00, CH_3_	6
14	115.14, CH_2_	115.78, CH_2_	1, 9	115.68, CH_2_	1, 9
15	15.04, CH_3_	15.07, CH_3_		15.12, CH_3_	

* Atom number referred to Figure 2; ^a^ Spectra recorded on a 125.77 MHz instrument; ^b^ assignments aided by APT and 2D NMR experiments (see the experimental section for further details).

**Table 3 molecules-27-07297-t003:** Allelopathic effect of *X. spinosum* root extracts on (**a**) *Allium porrum* (monocot plant) and (**b**) *Raphanus sativus* (dicot plant).

(**a**) *A. porrum*								
**Treatment**	**[C] (µg/mL)**	**L (mm) ***	**GR**	**GP**	**VI**	**LR**	**WR**	**AE**
EO	0 (Control)	34.2 ± 18.8 **a**	2.1	86.7	3.0	1.2	0.2	-
	100	14.1 ± 10.0 **b**	1.8	83.3	1.2	1.1	0.2	−58.8
	250	5.6 ± 6.5 **c**	1.2	73.3	0.4	1.3	0.2	−83.5
	500	4.9 ± 4.4 **c**	1.1	76.7	0.4	1.3	0.2	−85.8
	1000	3.7 ± 3.2 **c**	0.9	66.7	0.2	1.2	0.4	−89.2
HYD	0 (Control)	29.0 ± 20.7 **a**	2.2	80.0	2.3	1.2	0.2	-
	100	15.3 ± 10.7 **b**	1.8	83.3	1.3	1.2	0.2	−47.2
	250	7.5 ± 7.4 **b,c**	1.2	66.7	0.5	1.1	0.1	−73.4
	500	6.9 ± 6.1 **bc**	1.7	83.3	0.6	1.0	0.1	−76.1
	1000	5.3 ± 7.6 **c**	0.9	60.0	0.3	1.6	0.2	−81.6
MAE-F10	0 (Control)	48.6 ± 20.6 **a**	2.6	93.3	4.5	1.2	0.2	-
	100	30.9 ± 21.8 **b**	2.2	83.3	2.6	1.3	0.2	−36.4
	250	25.2 ± 14.8 **b**	2.0	83.3	2.1	1.4	0.1	−48.2
	500	4.2 ± 7.8 **c**	0.9	40.0	0.2	1.9	0.3	−91.3
	1000	2.1 ± 4.5 **c**	0.7	33.3	0.1	1.9	0.3	−95.7
(**b**) *R. sativus*								
**Treatment**	**[C] (µg/mL)**	**L (mm)***	**GR**	**GP**	**VI**	**LR**	**WR**	**AE**
EO	0 (Control)	147.3 ± 56.2 **a**	6.2	96.7	14.2	0.5	0.1	-
	100	137.6 ± 56.5 **a**	6.5	96.7	13.3	0.5	0.1	−6.6
	250	135.5 ± 53.3 **a**	6.0	96.7	13.1	0.7	0.1	−8.0
	500	126.4 ± 36.8 **a**	5.9	100.0	12.6	0.6	0.1	−14.2
	1000	125.1 ± 37.9 **a**	5.5	100.0	12.5	0.6	0.1	−15.1
HYD	0 (Control)	147.3 ± 56.2 **a**	6.2	96.7	14.2	0.5	0.1	-
	100	154.4 ± 45.7 **a**	6.0	100.0	15.4	0.5	0.1	4.9
	250	134.4 ± 51.9 **a,b**	5.4	100.0	13.4	0.5	0.1	−8.8
	500	121.7 ± 47.3 **b**	4.9	96.7	11.8	0.8	0.1	−17.4
	1000	80.6 ± 37.7 **c**	5.3	100.0	8.1	1.0	0.1	−45.3
MAE-F10	0 (Control)	158.3 ± 50.6 **a**	8.5	100.0	15.8	0.4	0.1	-
	100	152.0 ± 48.7 **a,b**	7.9	100.0	15.2	0.4	0.1	−4.0
	250	127.2 ± 57.8 **b,c**	8.0	100.0	12.7	0.6	0.1	−19.7
	500	96.1 ± 53.5 **c**	7.3	100.0	9.6	0.6	0.1	−39.3
	1000	114.7 ± 39.2 **c**	7.3	100.0	11.5	0.6	0.1	−27.6

C: treatment concentration (µg/mL); L: mean of the total length (mm), *: means within treatment row followed by the same letter (**a**, **b** or **c**) are not significantly different at *p* = 0.05 level according to Tukey test; GR: germination rate; GP: germination percentage (%); VI: vigor index; LR: lengths ratio; WR: weights ratio; AE: allelopathic effect (%). See Experimental section for further details.

## Data Availability

The study did not report any data.

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
