# Peer review of "Guaianolide Derivatives from the Invasive Xanthium spinosum L.: Evaluation of Their Allelopathic Potential"

_molecules, 2022, doi:10.3390/molecules27217297_

Round 1
Reviewer 1 Report
The researchers studied the composition of the essential oils from the roots of Xanthium spinosum for the first time, by conducting an array of experiments including column chromatography, GC-FID, GC-MS, NMR and hemi-synthesis. It is important to point out that the correct assignment or the complete record of NMR data of ziniolide derivatives 65–67 were reported. Moreover, the allelopathic potential of the essential oil and extracts were detected, indicating ziniolide derivatives might be crucial for invasion. Based on these findings, this manuscript will gain interests from readers. However, revisions are necessary before this manuscript was accepted.
Comments:
1: What are the criteria for judging the main compounds of EO? As shown in Table 1, the portions of α-humulene 38, β-selinene 42 and α-selinene 43 were 3.6 %, 3.4 % and 3.1 %, all of which were higher than xantholide B 65 (3.0 %), but they were not regarded as main compounds.
2: As shown in Table 1, the contents of compounds 65–67 in HYD were higher than MAE extract. Why was the HYD not selected as the source for separation?
3: The mass data ‘m/z 232’ was different from that displayed in Figure S1, and the molecular formula was not consistent with the structure 66 shown in Figure 2.
4: As the correct assignments of NMR data of 66 and the structure elucidation of 65–67 were supported by the HMBC experiment, it is necessary to add a figure in the manuscript to show the key HMBC correlations. In addition, revise ‘H-15’, ‘H-9’, ‘H-14’ and ‘H-13’ (Lines 167 & 168) as ‘H3-15’, ‘H2-9’, ‘H2-14’ and ‘H2-13’, respectively.
5: To establish the cis-stereochemistry of ring junction of 66 and the α orientation of Me-13 of 65, it is more convincing to perform and analyze the ROESY spectrum.
6: Please provide the NMR spectra of compounds 65–67 as Supplementary Materials.
7: What did the letters a, b, c after the numbers represent? Please specify it.
Others:
1. Please delete the dot in the Title.
2. P1L9, P15L511: ‘12,8-Guaianolide’ → ‘12,8-guaianolide’
3. P1L12: ‘Column chromatography’ → ‘column chromatography’
4. Please give the full name for the abbreviation ‘MAE’ when it appeared for the first time.
5. P1L32: ‘pathogenic organism’ → ‘pathogenic organisms’
6. P3L116: ‘monoterpenoïds’ → ‘monoterpenoids’
To keep consistence, please revise ‘D-Germacrene’ in Table 1 as ‘Germacrene D’ as shown in P3L116.
7. P6L153: ‘66 MS-spectrum’ → ‘The MS-spectrum of 66’
8. P7L180: ‘65 MS-spectrum’ → ‘The MS-spectrum of 65’
9. P7L185: ‘show’ → ‘showed’
10. P7L191&L195, P9L225&L232: ‘methyle’ → ‘methyl’
11. Please rewrite the sentence ‘which allowed by thorough NMR analysis an easy extraction of the fifteen chemical 220 shifts of the main compound 67.’
12. P9L225: ‘H-13’ → ‘H3-13’
13. P12L334: ‘Ziniolide 66 biological effect’ → ‘the biological effect of ziniolide 66’
14. P15L513: ‘was’ → ‘were’
15. Please revise the references according to the Instructions for Authors.
Author Response
Dear reviewer,
Many thanks for your comments. As requested, we have revised the manuscript according your recommendations. We provide as follow a point-by-point response.
Best regards
Alain Muselli
Review 1
Reponses of comments:
1: What are the criteria for judging the main compounds of EO? As shown in Table 1, the portions of α-humulene 38, β-selinene 42 and α-selinene 43 were 3.6 %, 3.4 % and 3.1 %, all of which were higher than xantholide B 65 (3.0 %), but they were not regarded as main compounds.
Main components were revised to only include the first five. The sentence was revised in the manuscript as follow : “The main components were α-isocomene 32 (6.1 %), β-elemene 33 (8.5 %), neryl 2-methylbutyrate 53 (6.2 %), carotol 55 (9.4 %) and ziniolide 66 (19.3 %).” The figure 1 was revised adding only the 5 above constituants.
2: As shown in Table 1, the contents of compounds 65–67 in HYD were higher than MAE extract. Why was the HYD not selected as the source for separation?
The HYD could have been selected to ensure separation and identification of compounds 65–67, but the MAE was more suitable, cause of his higher yield (0.23 %) which allowed further investigations including allelopathy. This detail is now specified (section 2.2), and not only indicated in the experimental section. Moreover, microwave extraction (MAE) was faster (20min extraction + 40min cooling) than hydrodistillation (5h) and require less steps than hydrosol extraction.
3: The mass data ‘m/z 232’ was different from that displayed in Figure S1, and the molecular formula was not consistent with the structure 66 shown in Figure 2.
This comment highlighted a bad confusion on ziniolide molecular mass and formula in the manuscript, and was granted with a prompt correction: m/z 230 C15H18O2.
4: As the correct assignments of NMR data of 66 and the structure elucidation of 65–67 were supported by the HMBC experiment, it is necessary to add a figure in the manuscript to show the key HMBC correlations. In addition, revise ‘H-15’, ‘H-9’, ‘H-14’ and ‘H-13’ (Lines 167 & 168) as ‘H3-15’, ‘H2-9’, ‘H2-14’ and ‘H2-13’, respectively.
HMBC spectrum of 66 and 67 were added in Supplementary materials. RMN 2D of 65 was not recorded.
5: To establish the cis-stereochemistry of ring junction of 66 and the α orientation of Me-13 of 65, it is more convincing to perform and analyze the ROESY spectrum.
We considered that NOESY or ROESY experiments of 65 and 66 were not very informative because both compounds were never isolated as pure. The cis-stereochemistry of ring junction of 66 and the a-orientation of the Me-13 of 65 were clearly admitted in the literature. So, comparison of the NMR 1D data from literature allowed the non-ambiguous identification of 65 and 66. Consequently, their stereochemistry were ensured. Concerning the b-derivative, the NaBH4 reduction of 66 produced a mixture of both 65 and 67; the compared analysis of both sets of NMR data allowed to identify 66 as b-dihydroziniolide.
6: Please provide the NMR spectra of compounds 65–67 as Supplementary Materials.
All the missing spectra were added in Supplementary materials.
7: What did the letters a, b, c after the numbers represent? Please specify it.
Concerning the interpretation of allelopathic assay, we chose to adopt statistical data treatment commonly used in the literature. Statistical analysis is often performed in studies as https://doi.org/10.3390/plants11081003;https://www.scielo.cl/pdf/gbot/v72n2/07.pdf;https://www.nature.com/articles/s41598-022-16203-5. As specified in the table footnote: “*: means within treatment row followed by the same letter are not significantly different at p = 0.05 level according to Tukey test”. More details are available in the experimental section (statistical analysis L511): “Seedlings lengths data from allelopathic bioassays was subjected to one-way analysis of variance (ANOVA) using Minitab statistical software. Means of multiple treatments were compared using Tukey HSD (Honestly Significant Difference) test at 5 % level of significance. All results are expressed as mean ± SD. *Means sharing a common letter are not significantly different.”
Others:
- Please delete the dot in the Title: OK
- P1L9, P15L511: ‘12,8-Guaianolide’ → ‘12,8-guaianolide’: OK
- P1L12: ‘Column chromatography’ → ‘column chromatography’: OK
- Please give the full name for the abbreviation ‘MAE’ when it appeared for the first time. : OK
- P1L32: ‘pathogenic organism’ → ‘pathogenic organisms’: OK
- P3L116: ‘monoterpenoïds’ → ‘monoterpenoids’: OK
To keep consistence, please revise ‘D-Germacrene’ in Table 1 as ‘Germacrene D’ as shown in P3L116. : OK
- P6L153: ‘66MS-spectrum’ → ‘The MS-spectrum of 66’: OK
- P7L180: ‘65MS-spectrum’ → ‘The MS-spectrum of 65’: OK
- P7L185: ‘show’ → ‘showed’: OK
- P7L191&L195, P9L225&L232: ‘methyle’ → ‘methyl’: OK
- Please rewrite the sentence ‘which allowed by thorough NMR analysis an easy extraction of the fifteen chemical 220 shifts of the main compound 67: The sentence was revised as follow in the manuscript: “The ultra-dominant abundance of 67 allowed extracting the fifteen chemical shifts from the 13C-NMR spectra”.
- P9L225: ‘H-13’ → ‘H3-13’: OK
- P12L334: ‘Ziniolide 66 biological effect’ → ‘the biological effect of ziniolide 66’: OK
- P15L513: ‘was’ → ‘were’: OK
- Please revise the references according to the Instructions for Authors.: The references were revised.

Reviewer 2 Report
Paper is well-written and describes interesting chemical evaluation of Xanthium spinosum. I must congratulate good work!
Some minor corrections: 1. Instead of D-Germacrene please write Germacrene-D; 2. cis, trans in names with italic; 3. Correct " D-Germacrene" 4. It is not clear, why some components of EO are not presented (or are not visible ) in hexane (MAC or MAE). All constituents of EOs are well solubile in hexane; 5. RI presented in Adam's database were recorded on HP-5 -type column, whereas authors used RTX-1. The differences are sometimes higher than 20, but in Table 1 data are almose identical. For example RI on HP-1 (lit.) for geraniol is 1235, in manuscript 1244 lit. and theoretical, wharead in Adams 1249. Please re-check what database was used for RI comparision. I recommend add crude chromatograms to supplementary data.
Author Response
Authors Revisions
Manuscript ID : molecules-1919878
Title : Guaianolide derivatives from the invasive Xanthium spinosum L. Evaluation of their allelopathic potential.
Dear reviewer,
Many thanks for your comments. As requested, we have revised the manuscript according your recommendations. We provide as follow a point-by-point response.
Best regards
Alain Muselli
Review 2
Some minor corrections:
- Instead of D-Germacrene please write Germacrene-D; OK
- cis, trans in names with italic; OK
- Correct " D-Germacrene" OK
- It is not clear, why some components of EO are not presented (or are not visible) in hexane (MAC or MAE). All constituents of EOs are well solubile in hexane; We assume that monoterpene were present in very small proportion (< 0.1 %) in MAC and MAE extracts, leading to a lack of integration. In addition, numerous of these compounds were not detected in the hydrocarbon fraction obtained by column chromatography from the bulk MAC and MAE extracts.
- RI presented in Adam's database were recorded on HP-5 -type column, whereas authors used RTX-1. The differences are sometimes higher than 20, but in Table 1 data are almose identical. For example RI on HP-1 (lit.) for geraniol is 1235, in manuscript 1244 lit. and theoretical, wharead in Adams 1249. Please re-check what database was used for RI comparision. I recommend add crude chromatograms to supplementary data. Some mistakes were corrected in Table 1 and RI were revised according to the following exigencies. Primarily, RI data from the Massfinder library were used as reference because GC experimental conditions were close to those of our lab. When Massfinder database was not applicable, Adam’s database was consulted. In addition, crude chromatograms of the four extracts were added in Supplementary material.

Round 2
Reviewer 1 Report
All the comments have been addressed and the manuscript is revised accordingly. Only the typo errors need to be revised, including ‘RMN 13C’ → ‘13C NMR’, ‘RMN 1H’ → ‘1H NMR’, ‘RMN HMBC’ → ‘HMBC’, ‘RMN HSQC’ → ‘HSQC’, ‘RMN COSY’ → ‘COSY’, ‘RMN NOESY’ → ‘NOESY’, but these could be corrected during the proof process. Thus this manuscript could be accepted in the present form.